biophysics/ecology/biomathematics

mean first passage time, Lotka–Volterra equations, invasion, niche overlap, mean time to extinction, demographic stochasticity

**Author for correspondence:**
Anton Zilman
e-mail: zilmana@physics.utoronto.ca

# Effects of niche overlap on coexistence, fixation and invasion in a population of two interacting species

Matthew Badali[1] and Anton Zilman[1,2]

[1]Department of Physics, University of Toronto, 60 St George St., Toronto, Canada M5S 1A7
[2]Institute for Biomaterials and Biomedical Engineering, University of Toronto, Toronto, Canada

 MB, 0000-0002-1198-2770

Synergistic and antagonistic interactions in multi-species populations—such as resource sharing and competition—result in remarkably diverse behaviours in populations of interacting cells, such as in soil or human microbiomes, or clonal competition in cancer. The degree of inter- and intra-specific interaction can often be quantified through the notion of an ecological 'niche'. Typically, weakly interacting species that occupy largely distinct niches result in stable mixed populations, while strong interactions and competition for the same niche result in rapid extinctions of some species and fixations of others. We investigate the transition of a deterministically stable mixed population to a stochasticity-induced fixation as a function of the niche overlap between the two species. We also investigate the effect of the niche overlap on the population stability with respect to external invasions. Our results have important implications for a number of experimental systems.

## 1. Introduction

Remarkable biodiversity exists in biomes such as the human microbiome [1–3], the ocean surface [4,5], soil [6], the immune system [7–9] and other ecosystems [10,11]. Accordingly, quantitative predictive understanding of the long-term population behaviour of complex populations is important for many human health and disease and industrial processes such as drug resistance in bacteria, cancer progression, evolutionary phylogeny inference algorithms and immune response [2,3,12–17]. Nevertheless, the long-term dynamics, diversity and stability of communities of multiple interacting species are still incompletely understood.

One common theory, known as the Gause's rule or the competitive exclusion principle, postulates that due to abiotic constraints, resource usage, inter-species interactions and other factors, ecosystems can be divided into ecological niches, with each niche supporting only one species in steady state; that species is termed to have fixated within the niche [18–21]. Commonly, the number of ecological niches can be related to the number of limiting factors that affect growth and death rates, such as metabolic resources or secreted molecular signals like growth factors or toxins, or other regulatory molecules [22–25]. The strength of inter-species interactions can be commonly characterized by a quantity known as a niche overlap [26–30]. Observed biodiversity can also arise from the turnover of transient mutants or immigrants that appear and go extinct in the population [31–33]. However, the exact definition of an ecological niche is still a subject of debate and varies among different works [4,27,30,34–38]. The maintenance of the biodiversity of species that occupy similar niches is still not fully understood [25,39,40].

Deterministically, ecological dynamics of mixed populations has been commonly described as a dynamical system of equations that governs the dynamics of the numbers of individuals of each species and the concentrations of the limiting factors [22–24]. Steady-state coexistence of multiple species commonly corresponds to a stable fixed point in such a dynamical system, and the number of stably coexisting species is typically constrained by the number of limiting factors [22–24]. In some cases, deterministic models allow coexistence of more species than the number of limiting factors, for instance, when the attractor of the system dynamics is a limit cycle rather than a fixed point [24,41]. Particularly pertinent for this paper is the case when the interactions of the limiting factors and the target species have a redundancy that results in the transformation of a stable fixed point into a marginally stable manifold of fixed points. This situation underlies many neutral models of population dynamics and evolution [19,20,31,42–45]. However, in this situation, the stochastic fluctuations in the species numbers become important [24,25,46–49].

Stochastic effects, arising either from the extrinsic fluctuations of the environment [50,51], or the intrinsic stochasticity of the individual birth and death events within the population [38,43,52–58], modify the deterministic picture. The latter type of stochasticity, known as demographic noise, is the focus of this paper. Demographic noise causes fluctuations of the populations' abundances around the deterministic steady state until a rare large fluctuation leads to an extinction of one of the species [20,43,57]. In systems with a deterministically stable coexistence point, the mean time to extinction (MTE) is typically exponential in the population size [54,59–62] and is commonly considered to imply stable long-term coexistence for typical biological examples with relatively large numbers of individuals [62,63].

By contrast, in systems with a stable neutral manifold that restore fluctuations out of the manifold but not along it, mean extinction times scale as a power law with the population size, indicating that the coexistence fails in such systems on biologically relevant timescales [42,43,54,64,65]. This type of stochastic dynamics parallels the stochastic fixation in the Moran–Fisher–Wright model that describes strongly competing populations with fixed overall population size, and is the classical example of the class of models known as neutral models [9,14,20,37,42,66–68].

A broad class of dynamical models of multi-species populations interacting through limiting factors can be mapped onto the class of models known as generalized Lotka–Volterra (LV) models, which allow one to conveniently distinguish between various interaction regimes, such as competition or mutualism, and which have served as paradigmatic models for the study of the behaviour of interacting species [30,38,43,46–49,54–58,69,70]. Remarkably, the stochastic dynamics of LV type models is still incompletely understood, and has recently received renewed attention motivated by problems in bacterial ecology and cancer progression [9,25,37,38,57,71–73]. A number of approximations have been used to understand the stochastic dynamics of the LV model, predominantly in the independent regime, or the opposite neutral limit [38,43,54–57,74]. However, the approximations tend to break down for the strong competition near the neutral limit, which is of central importance for understanding the deviations of ecological and physiological systems from neutrality [38,70,75–77].

In this paper, we analyse a LV model of two competing species using the master equation and first passage formalism that enables us to obtain numerically exact solution to arbitrary accuracy in all regimes, avoiding inaccuracies of various approximate descriptions of the stochastic dynamics of the system. The emphasis of this paper is on the effect of the ecological niche overlap on the transition from the deterministic coexistence to the stochastic fixation. In particular, we focus on the balance between the stochastic and deterministic effects in the strong competition regime near-complete niche overlap, as expressed in the interplay between the exponential and algebraic terms in the extinction time as a function of the population size. Furthermore, we study the complementary question of the population stability with respect to the invasion of an immigrant or a mutant into a stable ecological

niche of an already established species. This analysis informs many theories of transient diversity, such as Hubbell's neutral theory, and serves as a stepping stone towards multi-species models in non-neutral regime.

The paper is structured as follows. Section 2 discusses the definition of an ecological niche and briefly examines the regimes of deterministic stability of the system. In §3, we introduce the stochastic description of the LV model. In §3.1, we analyse the fixation/extinction times as a function of the niche overlap between the two species. We find that for any incomplete niche overlap the extinction time contains both terms that scale exponentially and algebraically with the system size, only reaching the known algebraic scaling at complete niche overlap; we calculate the exact functional form of these dependencies. Section 3.2 discusses the dynamic underpinning of the results of §3.1. In §3.3, we consider the invasion of an immigrant or a mutant into a stable ecological niche of an already established species, and calculate the invasion probabilities as a function of the system size and the niche overlap. The probability of success of an invasion attempt depends strongly on the competition strength, decreasing linearly with niche overlap, while only weakly dependent on the system size. All invasion attempts occur rapidly regardless of the outcome, or of the niche overlap between the two species or system size. We conclude with the Discussion section of these results in the context of previous works, and their implications for a number of experimental systems.

# 2. Deterministic behaviour of the Lotka–Volterra model: overview

The model being employed in this paper is based on the generalized two-dimensional LV equations, that have served as a paradigmatic system to study the behaviour of interacting populations [30,38,43,46–49,54–58,69,70]. An example of the derivation of the model, from the underlying mechanistic model describing the exchange of control factors, is shown in the electronic supplementary material. The dynamical equations of the numbers of individuals of species 1 (denoted as $x_1$) and species 2 (denoted as $x_2$) are [57,78,79]

$$\dot{x}_1 = r_1 x_1 \left(1 - \frac{x_1 + a_{12}x_2}{K_1}\right) \quad \text{and} \quad \dot{x}_2 = r_2 x_2 \left(1 - \frac{a_{21}x_1 + x_2}{K_2}\right). \tag{2.1}$$

The turnover rates $r_i$ set the timescales of the birth and death for each species, and $K_i$ are known as the carrying capacities. The interaction parameters $a_{ij}$ provide a mathematical representation of the intuitive notion of the niche overlap between the species, commonly used to characterize the competition of multi-species systems. These niche overlap parameters, which represent the competition between species arising from their shared usage of resources and other factors, dictate the long-term coexistence, viability and the diversity of the population, as well as the apportionment of the different species among different niches [26–30]. The niche overlap parameters can be directly derived from an underlying model describing the production–consumption dynamics of the limiting factors that control the birth and death rates in the system (see electronic supplementary material). When $a_{ij} = 0$, species $j$ does not affect species $i$, and they occupy separate ecological niches. At the other limit, $a_{ij} = 1$, species $j$ competes just as strongly with species $i$ as species $i$ does within itself, and both species occupy same niche.

The deterministic behaviour of the generalized LV model is well known and can result in various stability regimes, including coexistence, competitive exclusion, mutualism and bi-stability [57,78,79], as reviewed in the electronic supplementary material. Here, we briefly review the behaviour of the deterministic equations (2.1), which have four fixed points [57,78,79]

$$O = (0, 0), \quad A = (0, K_2), \quad B = (K_1, 0) \quad \text{and} \quad C = \left(\frac{K_1 - a_{12}K_2}{1 - a_{12}a_{21}}, \frac{K_2 - a_{21}K_1}{1 - a_{12}a_{21}}\right). \tag{2.2}$$

The origin $O$ is the fixed point corresponding to both species being extinct, and is unstable with positive eigenvalues equal to $r_1$ and $r_2$ along the corresponding on-axis eigendirections. The single species fixed points $A$ and $B$ are stable on axis (with eigenvalues $-r_1$ and $-r_2$, respectively), but are unstable with respect to invasion if point $C$ is stable, reflected in the positive second eigenvalue equal to $r_2(1 - a_{21}K_1/K_2)$ and $r_1(1 - a_{12}K_2/K_1)$, respectively. Fixed point $C$ corresponds to the coexistence of the two species and is stable when both $a_{ij} < 1$ for $K_1 = K_2$, also known as the weak competition regime ([57,78,79] and electronic supplementary material).

In the special case $a_{12} = a_{21} = 1$, the stable coexistence point degenerates into a neutral line of stable points, defined by $x_2 = K - x_1$, as shown in figure 1. Each point on the line is stable with respect to

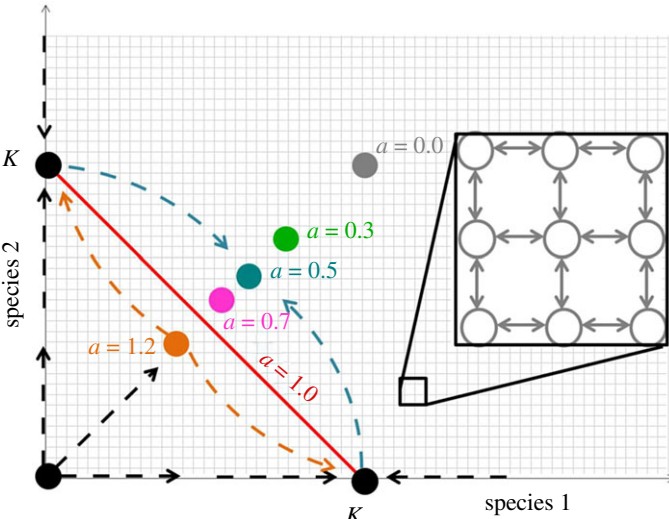

**Figure 1.** Phase space of the coupled logistic model. Coloured dots show the position of $C = ((K_1 - a_{12}\, K_2)/(1 - a_{12}a_{21})$, $(K_2 - a_{21}\, K_1)/(1 - a_{12}a_{21}))$ at the indicated values of the niche overlap $a$ for $a_{12} = a_{21} \equiv a$. The fixed point is stable for $a < 1$. At $a = 0$, the two species evolve independently. As $a$ increases, the deterministically stable fixed point moves towards the origin. At $a = 1$, the fixed point degenerates into a line of marginally stable fixed points, corresponding to the Moran model. The dashed lines illustrate the deterministic flow of the system: blue for $a = 0.5$, and orange for $a = 1.2$. The zoom inset illustrates the stochastic transitions between the discrete states of the system. Fixation occurs when the system reaches either of the axes. See text for details.

perturbations off line, but any perturbations along the line are not restored to their unperturbed position [23,80]. This line corresponds to the singular case of complete niche overlap where the two species are functionally identical with respect to their interactions with limiting factors like resources, space, toxins, predators etc. (see also electronic supplementary material). The stochastic dynamics along this line correspond to the classical Moran model [43,56–58] as discussed below, and in the following we refer to this line as the Moran line.

Figure 1 shows the phase portrait of the system, in the symmetric case of $K_1 = K_2 \equiv K$, $r_1 = r_2 \equiv r$ and $a_{12} = a_{21} \equiv a$, where neither of the species has an explicit fitness advantage. This equality of the two species, also known as neutrality, serves as a null model against which systems with explicit fitness differences can be compared.

Stochastic effects, described in the next section, modify the deterministic stability picture and can lead to extinction or fixation of species even in deterministically stable case. In this paper, we focus on the effects of stochasticity in the deterministically stable weak competition regime ($0 \leq a \leq 1$), finding the scaling of the mean times to fixation and invasion as a function of the niche overlap $a$ and comparing these to the known limits. The results in the asymmetric case are qualitatively similar and are relegated to the electronic supplementary material.

# 3. Effects of stochasticity

Stochasticity naturally arises in the dynamics of the system from the randomness in the birth and death times of the individuals—commonly known as the demographic noise [52,81–83]. Competitive interactions between the species can affect either the birth rates (such as competition for nutrients) or the death rates (such as toxins or metabolic waste), and in general may result in different stochastic descriptions [84,85]. In this paper, we follow others [43,55,56] in considering the case where the inter-species competition affects the death rates so that the *per capita* birth and death rates $b_i$ and $d_i$ of species $i$ are

$$\frac{b_i}{x_i} = r_i \quad \text{and} \quad \frac{d_i}{x_i} = r_i \frac{x_i + a_{ij}x_j}{K_i}. \tag{3.1}$$

The *per capita* death rate of a member of species $i$ is increased by the presence of other members of the same species, and to a lesser effect (in proportion to their niche overlap) also by members of the other

species. In the deterministic limit of negligible fluctuations, the model recovers the mean field competitive LV equations (2.1) [43]. The effects of an intrinsic death rate and competition modifying birth rates will be studied elsewhere.

The system is characterized by the vector of probabilities $P(s, t \mid s^0)$ to be in a state $s = \{x_1, x_2\}$ at time $t$, given the initial conditions $s^0 = (x_1^0, x_2^0)$: $[\boldsymbol{P}](t) \equiv (\cdots, P(s, t \mid s^0), \cdots)$ [86]. The forward master equation describing the time evolution of this probability distribution is [81]

$$\frac{\mathrm{d}}{\mathrm{d}t}[\boldsymbol{P}](t) = \hat{M}[\boldsymbol{P}](t), \tag{3.2}$$

where $\hat{M}$ is the (semi-infinite) transition matrix, with more details given in the following paragraph.

Because the approximate analytical and semi-analytical solutions of the master equation (3.2) often do not provide correct scaling in all regimes ([85,87–89]; see also the electronic supplementary material), we analyse the master equation numerically in order to recover both the exponential and polynomial aspects of the mean time to fixation. To enable numerical manipulations, we introduce a reflecting boundary condition at a cut-off population size $C_K > K$ for both species to make the transition matrix finite [86,90,91] and enumerate the states of the system with a single index [86] via the mapping of the two species populations $(x_1, x_2)$ to state $s$ as

$$s(x_1, x_2) = (x_1 - 1)C_K + x_2 - 1, \tag{3.3}$$

where $s$ serves as the index for our concatenated probability vector.

In this representation, the non-zero elements of the sparse matrix $\hat{M}$ are $\hat{M}_{s,s} = -b_1(s) - b_2(s) - d_1(s) - d_2(s)$ along the diagonal, giving the rate of transition out of state $(x_1, x_2)$ to the adjacent states $(x_1 + 1, x_2)$, $(x_1, x_2 + 1)$, $(x_1 - 1, x_2)$ or $(x_1, x_2 - 1)$, respectively, corresponding to a species 1 or 2 birth or death event. Off the diagonal, the elements $\hat{M}_{s,s+1} = d_2(s + 1)$ give the transition rate from the state of populations $(x_1, x_2)$ of the two species to the state $(x_1, x_2 - 1)$ organisms, corresponding to the death of a member of species 2. Off-diagonal elements $\hat{M}_{s+1,s} = b_2(s)$ are the transition from state $(x_1, x_2)$ to state $(x_1, x_2 + 1)$, similarly corresponding to the birth of an organism from species 2. The off-diagonal elements at $\pm C_K$ are the remaining two transitions: the death of species 1 member is given by $\hat{M}_{s,s+C_K} = d_1(s + C_K)$, and its birth is $\hat{M}_{s+C_K,s} = b_1(s)$. Some diagonal elements are modified to ensure the reflecting boundary at $x_i = C_K$. We have found that the choice $C_K = 5K$ is more than sufficient to calculate the mean fixation times to at least three significant digits of accuracy.

## 3.1. Fixation time as a function of the niche overlap

In this section, we calculate the first passage times to the extinction of one of the species and the corresponding fixation of the other, induced by demographic fluctuations, starting from an initial condition of the deterministically stable coexistence point. The stochastic system tends to fluctuate near the deterministic fixed point (or Moran line), but rare fluctuations can take the system far from equilibrium. Should the fluctuations bring the system to either of the axes, then one species has gone extinct. In this model, each axis acts as an absorbing manifold, from which the system cannot recover. Biologically, once a species has gone extinct from the system it can no longer reproduce. The system then proceeds to evolve as a one species logistic model with demographic noise, a process that is well studied in the literature [59,62,92].

The time the system takes to fixation is a random variable, the distribution of which can be calculated in a number of ways. The master equation (3.2) has a formal solution obtained by the exponentiation of the matrix: $[\boldsymbol{P}](t) = \mathrm{e}^{\hat{M}t}[\boldsymbol{P}](0)$. Then $([\boldsymbol{P}])_{s=(x_1,0)}$ and $([\boldsymbol{P}])_{s=(0,x_2)}$ give the cumulative distribution of the fixation of species 1 and 2, respectively. However, direct matrix exponentiation, as well as direct sampling of the master equation using the Gillespie algorithm [93,94], are impractical, since the fixation time grows exponentially with the system size; nevertheless, we used Gillespie tau-leaping simulations to verify our results up to moderate system size (see electronic supplementary material). More amenably, the moments of the first passage time distribution can be calculated directly without explicitly solving the master equation [95]. In an ensemble of trajectories, the fraction of trajectories in state $s$ at time $t$ is given by $P(s, t \mid s^0)$. Thus, averaging over the ensemble, the mean residence time in state $s$ during the system evolution is the integral of this quantity over all time [95]

$$\langle t(s^0) \rangle_s = \int_0^\infty \mathrm{d}t \, P(s, t \mid s^0) = \int_0^\infty \mathrm{d}t \, (\mathrm{e}^{\hat{M}t})_{s,s^0} = -(\hat{M}^{-1})_{s,s^0}. \tag{3.4}$$

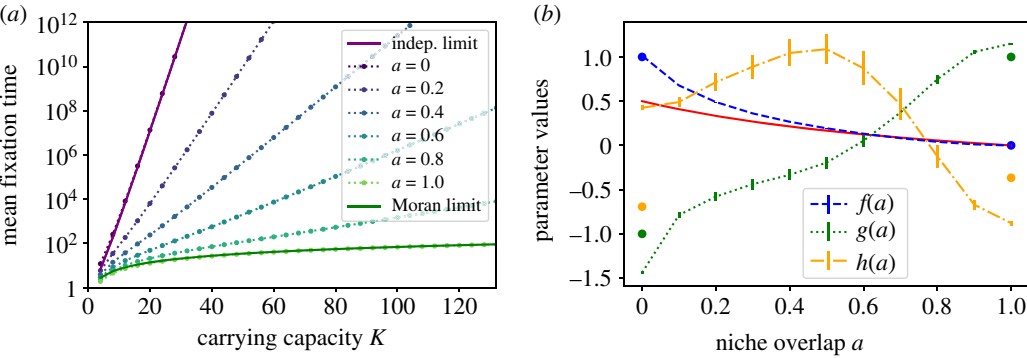

**Figure 2.** (a) Dependence of the fixation time on carrying capacity and niche overlap. The lowest line, $a = 1$, recovers the Moran model results with the fixation time algebraically dependent on $K$ for $K \gg 1$. For all other values of $a$, the fixation time is exponential in $K$ for $K \gg 1$. (b) Niche overlap controls the transition from coexistence to fixation. Blue line: $f(a)$ from the ansatz of equation (3.6) characterizes the exponential dependence of the fixation time on $K$; it smoothly approaches zero as the niche overlap reaches its Moran line value $a = 1$. Green line: $g(a)$ quantifies the scaling of the pre-exponential prefactor $K^{g(a)}$ with $K$. Yellow line: $h(a)$ is the multiplicative constant. The bars represent a 95% confidence interval, and for $f$ are thinner than the line. The dots at the extremes $a = 0$ and $a = 1$ are the expected asymptotic values. Red line: Gaussian approximation to Fokker–Planck equation (see next subsection).

For each trajectory, the time to reach fixation is simply the sum of the times it spends in each state on the way. Thus the mean time to fixation starting from a state $s^0$ is the sum of the residency times [96]

$$\tau(s^0) = -\sum_s \langle t(s^0) \rangle_s = -\sum_s (\hat{M}^{-1})_{s,s^0}. \tag{3.5}$$

This expression can also be derived using the backward master equation formalism [96]. The matrix inversion was performed using LU decomposition algorithm implemented with the C++ library Eigen [97], which has algorithmic complexity of the calculation scaling algebraically with $K$. Increasing the cut-off $C_K$ enables calculation of the mean fixation times to an arbitrary accuracy.

Figure 2a shows the calculated fixation times with the initial condition at the deterministically stable coexistence fixed point as a function of the carrying capacity $K$ for different values of the niche overlap $a$. While the transition has not previously been characterized, the independent and neutral limits are known, providing comparisons at the $a = 0$ and $a = 1$ extremes.

In the limit of non-interacting species ($a = 0$), each species evolves according to an independent stochastic logistic model, and the probability distribution of the fixation times is a convolution of the extinction time distributions of a single species, which are dominated by a single exponential tail [59,62,92] (see also the electronic supplementary material). Mean extinction time of a single species can be calculated exactly and is well known, and asymptotically for $K \gg 1$ it varies as $1/Ke^K$ [98,99]. Denoting the probability distribution of the extinction times for either of the independent species starting from $K$ asymptotically for large $K$ as $p(t) = \alpha e^{-\alpha t}$ and its cumulative as $f(t) = \int_{s=0}^t p(s)\,\mathrm{d}s$, the probability that *either* of the species goes extinct in the time interval $[t, t + \mathrm{d}t]$, is $p_{\min}(t)\,\mathrm{d}t = (p(t)(1 - f(t)) + (1 - f(t))p(t))\,\mathrm{d}t$. Thus the mean time to fixation is $\langle t \rangle = \int_0^\infty \mathrm{d}t\, t\, p_{\min}(t) = 1/2\alpha$, where $1/\alpha = (1/K)e^K$ is the mean time of the single species extinction. This analytical limit $\tau \simeq (1/2K)e^K$ is shown in figure 2 as a solid purple line and agrees well with the numerical results of equation (3.5). From the biological perspective, for sufficiently large $K$, the exponential dependence of the fixation time on $K$ implies that the fluctuations do not destroy the stable coexistence of the two species.

In the opposite limit of the complete niche overlap, $a = 1$, any fluctuations along the line of neutrally stable points are not restored, and the system performs diffusion-like motion that closely parallels the random walk of the classic Moran model [38,43,49,54,56,57,74] (see also the electronic supplementary material). The Moran model shows a relatively fast fixation time scaling algebraically with $K$ [42,43], $\tau \simeq \ln(2)K^2\Delta t$, where $\Delta t$ is one-time step of a Moran model in our non-dimensionalized time units. This known result can be obtained from the backward Fokker–Planck equation $\partial^2 \tau / \partial x^2 = -\Delta t K^2 / x(K - x)$ of the Moran model, which directly gives the above scaling for equal population initial conditions [42,43]. The fixation time predicted by the Moran model is shown in figure 2 as a green solid line and shows excellent agreement with our exact result. Note that the average time step $\Delta t$ in the corresponding Moran model is $\Delta t \approx 1/K$ because the mean transition time in the stochastic LV model

is proportional to $1/(rK)$ close to the Moran line [57]; see the electronic supplementary material for more details. The relatively short fixation time in the complete niche overlap regime implies that the population can reach fixation on biologically realistic timescales.

The exponential scaling of the fixation time with $K$ persists for incomplete niche overlap described by the intermediate values of $0 < a < 1$. However, both the exponential and the algebraic prefactor depend on the niche overlap $a$. The exponential scaling is expected for systems with a deterministically stable fixed point [38,55,62,88,89], as indicated in [43,54,57] using Fokker–Planck approximation and in [55,100,101] using the Wentzel–Kramers–Brillouin (WKB) approximation. However, the Fokker–Planck and WKB approximations, while providing the qualitatively correct dominant scaling, do not correctly calculate the scaling of the polynomial prefactor and the numerical value of the exponent simultaneously [62,74,85]. For large population sizes and timescales, effective species coexistence will be typically observed experimentally whenever the fixation time has a non-zero exponential component.

To quantitatively investigate the transition from the exponentially stable fixation times to the algebraic scaling in the complete niche overlap regime, we use the ansatz

$$\tau(a, K) = e^{h(a)} K^{g(a)} e^{f(a)K}. \tag{3.6}$$

In the Moran limit, $a = 1$, we expect $f(1) = 0$, $g(1) = 1$. In the independent species limit with zero niche overlap, $a = 0$, we expect $f(0) = 1$ and $g(0) = -1$. Figure 2b shows the ansatz functions $f(a)$, $g(a)$ and $h(a)$, obtained by numerical fit to the fixation times as a function of $K$ shown in figure 2a.

The numerical results agree well with the expected approximate analytical results for $a = 0$ and $a = 1$ with small discrepancies attributable to the approximate nature of the limiting values. Notably, $f(a)$, which quantifies the exponential dependence of the fixation time on the niche overlap $a$, smoothly decays to zero at $a = 1$: only when two species have complete niche overlap ($a = 1$) does one expect rapid fixation dominated by the algebraic dependence on $K$. In all other cases, the mean time until fixation is exponentially long in the system size [62,92]. Even two species that occupy *almost* the same niche ($a \lesssim 1$) effectively coexist for $K \gg 1$, with small fluctuations around the deterministically stable fixed point; for a discussion of small $K$, see the next paragraph and electronic supplementary material. The dependence of the exponential function $e^{f(a)K}$ on $a$ and $K$ can be understood in the spirit of Kramers' theory, as discussed in the next section.

Interestingly, for small $K$ the algebraic prefactors may dominate the exponential scaling such that in a region of $a$ and $K$ space, the fixation time from a deterministically stable fixed point (with $a < 1$) can be shorter than that of a Moran case with the same $K$. See the electronic supplementary material for more details. Also explored in the electronic supplementary material is a breaking of the parameter symmetry, which gives similar behaviour to the above. In particular, the exponential dependence of the escape time from the fixed point on the system size also persists in the non-neutral case, although the dependence can be much weaker.

## 3.2. Route to fixation and the origin of the exponential scaling

To gain deeper insight into the fixation dynamics and to understand the origins and the limits of the exponential scaling with $K$, in this section, we calculate the residency times in each state during the fixation process, given by equation (3.4). The results are shown as a contour plot in figure 3, for two different niche overlaps, one at and the other far from the Moran limit. The set of states lying along the steepest descent lines of the contour plot, shown as the black line, can be thought of as a 'typical' trajectory [55,74,102]. However, even for two species close to complete niche overlap the system trajectory visits many states far from this line. This departure is even greater for weakly competing species, where the system covers large areas around the fixed point before the rare fluctuation that leads to fixation occurs [53]. These deviations from a 'typical' trajectory are related to the inaccuracy of the WKB approximation in calculating the scaling of the pre-exponential factor [53,89,99]; see also the electronic supplementary material. This occupancy landscape can be qualitatively thought of as an effective Lyapunov function/effective potential of the system dynamics [103], although the LV system does not possess a true Lyapunov function.

A similar issue also arises in the Fokker–Planck approximation [57,103]. Nevertheless, the pseudo-potential (also known as quasi-potential) landscape provides an intuitive underpinning for the general exponential scaling in the incomplete niche overlap regime: the fixation process can be thought of as the Kramers-type escape from a pseudo-potential well [104]. The Kramers' escape time is dominated by the exponential term $\tau \simeq \exp(\Delta U)$, where $\Delta U$ is the depth of the effective potential well, corresponding to the dominant scaling $\tau \simeq \exp(f(a)K)$ of this paper. The depth of the pseudo-potential

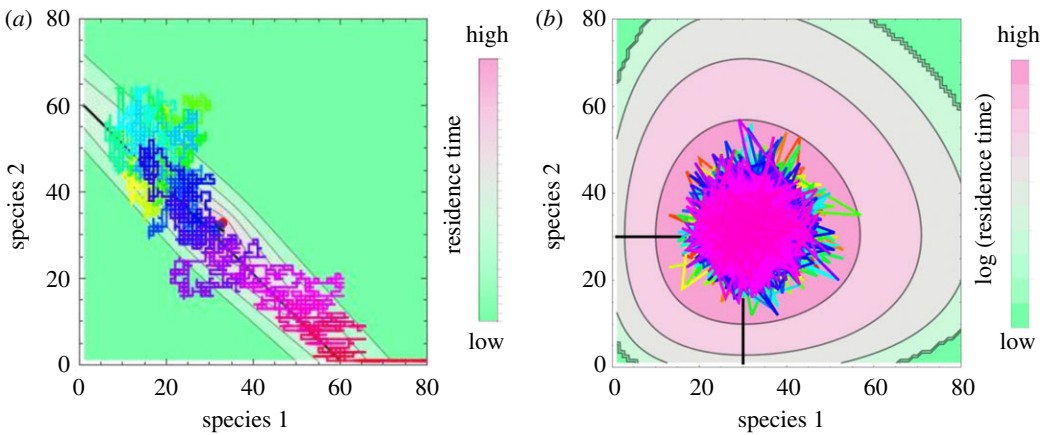

**Figure 3.** The system samples multiple trajectories on its way to fixation. Contour plot shows the average residency times at different states of the system, with pink indicating longer residence time, green indicating rarely visited states. The coloured line is a sample trajectory the system undergoes before fixation; colour coding corresponds to the elapsed time, with orange at early times, purple at the intermediate times and red at late stages of the trajectory. The red dot shows the deterministic coexistence point. See text for more details. (a) Complete niche overlap limit, $a = 1$, for $K = 64$. (b) Independent limit with $a = 0$ and $K = 32$.

well can be estimated in the Gaussian approximation to the Fokker–Planck equation [81,105,106]. For $y_i \equiv x_i/K$, the Fokker–Planck equation is $\partial_t P = -\sum_i \partial_i F_i P + (1/2K)\sum_{i,j} \partial_i \partial_j D_{ij} P$ with $F_1 = y_1(1 - y_1 - a\,y_2)$, $D_{11} = y_1\,(1 + y_1 + a\,y_2)$, similar terms for $F_2$ and $D_{22}$, and $D_{12} = D_{21} = 0$. Linearized about the deterministic fixed point $[y]^*$ the Fokker–Planck equation becomes $\partial_t P = -\sum_{i,j} A_{ij}\partial_i(y_j - y_j^*)P + (1/2K)\sum_{i,j} B_{ij}\partial_i\partial_j P$, where $A_{ij} = \partial_j F_i |_{[y]=[y]^*}$ and $B_{ij} = D_{ij}|_{[y]=[y]^*}$. The steady-state solution to this approximation is a Gaussian distribution centred on the deterministic fixed point. It gives an effective well depth of $\Delta U = ((1-a)/2(1+a))K$, providing a qualitative basis for the numerical results in figure 2b. As $a$ increases and the species interact more strongly, the potential well becomes less steep, resulting in weaker exponential scaling. In the complete niche overlap limit, the pseudo-potential develops a soft direction along the Moran line that enables relatively fast escape towards fixation. The change in the topology of the pseudo-potential is also reflected in the Pearson correlation coefficient between the two species: $\rho = \text{cov}(x_1, x_2)/\sqrt{\sigma_1^2 \sigma_2^2} = -a$. The covariance ranges from zero in the independent limit (as expected) to anti-correlated in the Moran limit, reflecting the fact that the trajectories typically diffuse along the trough around the Moran line.

## 3.3. Invasion of a mutant/immigrant into a deterministically stable population

Transient coexistence during the fixation/extinction process of immigrants/mutants has also been proposed as a mechanism for observed biodiversity in a number of contexts [31,35,70,73,75,107,108]. The extent of this biodiversity is constrained by the interplay between the residence times of these invaders and the rate at which they appear in a settled population. In the previous sections, we calculated the fixation times in the two-species system starting from the deterministically stable fixed point. In this section, we investigate the complementary problem of robustness of a stable population of one species with respect to an invasion of another species, arising either through mutation or immigration, and investigate the effect of niche overlap and system size on the probability and mean times of successful and failed invasions.

To this end, we study the case where the system starts with $K - 1$ individuals of the established species and 1 invader. Each species' dynamics is described by the birth and death rates defined by equations (3.1) above. We consider the invasion successful if the invader grows to be half of the total population without dying out first. Note that this is different from the bulk of invasion literature [43,49,54–58,64,70,90,107,109–112] (but see [113]), which typically considers invasion successful when a mutant fixates in the system. However, this is only sensible in the Moran limit, since otherwise the system tends to the coexistence fixed point, after which there is an equal chance of fixation and extinction, with the MTE calculated above; hence our definition of invasion. We denote the probability of a invader success as $\mathcal{P}$, the mean time to a successful invasion as $\tau_s$, and the mean time of a failed invasion attempt, where the invader dies out before establishing itself in the population, as $\tau_f$. More

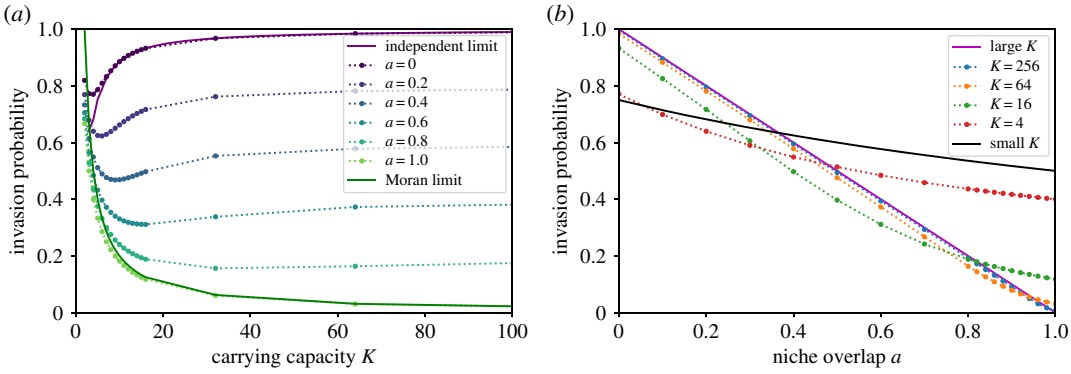

**Figure 4.** Probability of a successful invasion. (*a*) Solid lines show the numerical results, from $a = 0$ at the top to $a = 1$ at the bottom. The purple solid line is the expected analytical solution in the independent limit. The green solid line is the prediction of the Moran model in the complete niche overlap case. Numerical data are obtained using equation (3.7) and are connected with dotted lines to guide the eye. (*b*) The red dotted line shows the results for carrying capacity $K = 4$, the solid black line $\frac{b_{\mathrm{mut}}}{b_{\mathrm{mut}} + d_{\mathrm{mut}}}$ is an appropriate analytical approximation in the small carrying capacity limit. Successive lines are at larger system sizes, and approach the solid magenta line of $1 - d_{\mathrm{mut}}/b_{\mathrm{mut}} \approx 1 - a$ at very large $K$.

generally, invasion probability and the successful and failed times starting from an arbitrary state $s^0$ are denoted as $\mathcal{P}^{s^0}$, $\tau_s^{s^0}$ and $\tau_f^{s^0}$, respectively.

Similar to equation (3.5) above, the invasion probability can be obtained from [96,105]

$$\mathcal{P}^{s^0} = -\sum_s \hat{M}_{s,s^0}^{-1} \alpha_s \tag{3.7}$$

and the times from

$$\Phi^{s^0} = -\sum_s \hat{M}_{s,s^0}^{-1} \mathcal{P}^s, \tag{3.8}$$

where $\alpha_s$ is the transition rate from a state $s$ directly to extinction or invasion of the invader and $\Phi^{s^0} = \tau^{s^0} \mathcal{P}^{s^0}$ is a product of the invasion or extinction time and probability. Similar equations describe $\tau_f$ [96,105].

Figure 4 shows the calculated invasion probabilities as a function of the carrying capacity $K$ and of the niche overlap $a$ between the invader and the established species. In the complete niche overlap limit, $a = 1$, the dependence of the invasion probability on the carrying capacity $K$ closely follows the results of the classical Moran model, $\mathcal{P}^{s^0} = 2/K$ [42,90], shown in the solid green line in figure 4*a*, and tends to zero as $K$ increases. In the other limit, $a = 0$, the problem is well approximated by the one-species stochastic logistic model starting with one individual and evolving to either 0 or $K$ individuals; the exact result in this limit is shown as a black dotted line, referred to as the independent limit [105]. In the independent limit, $a = 0$, the invasion probability asymptotically approaches 1 for large $K$, reflecting the fact that the system is deterministically drawn towards the deterministic stable fixed point with equal numbers of both species. Interestingly, the invasion probability is a non-monotonic function of $K$ and exhibits a minimum at an intermediate/low carrying capacity, which might be relevant for some biological systems, such as in early cancer development [13] or plasmid exchange in bacteria [17]. The explanation for this non-monotonicity is that increasing $K$ has opposing effects: it makes the end goal of growing from one invader to half the population farther away, but it also increases the effective draw towards the deterministic fixed point.

For the intermediate values of the niche overlap, $0 < a < 1$, the invasion probability is a monotonically decreasing function of $a$, as shown in figure 4*b*. For large $K$, the outcome of the invasion is typically determined after only a few steps: since the system is drawn deterministically to the mixed fixed point, the invasion is almost certain once the invader has reproduced several times. At early times, the invader birth and death rates (3.1) are roughly constant, and the invasion failure can be approximated by the extinction probability of a birth–death process with constant death $d_{\mathrm{mut}}$ and birth $b_{\mathrm{mut}}$ rates. The invasion probability is then $\mathcal{P} = 1 - d_{\mathrm{mut}}/b_{\mathrm{mut}} \approx 1 - a$. This heuristic estimate is in excellent agreement with the numerical predictions, shown in figure 4*b* as a solid magenta and the dotted blue lines, respectively. For this reason, at large $K$, the dependence of the invasion probability on $a$ is much stronger than its dependence on $K$, such that increasing the carrying capacity has little

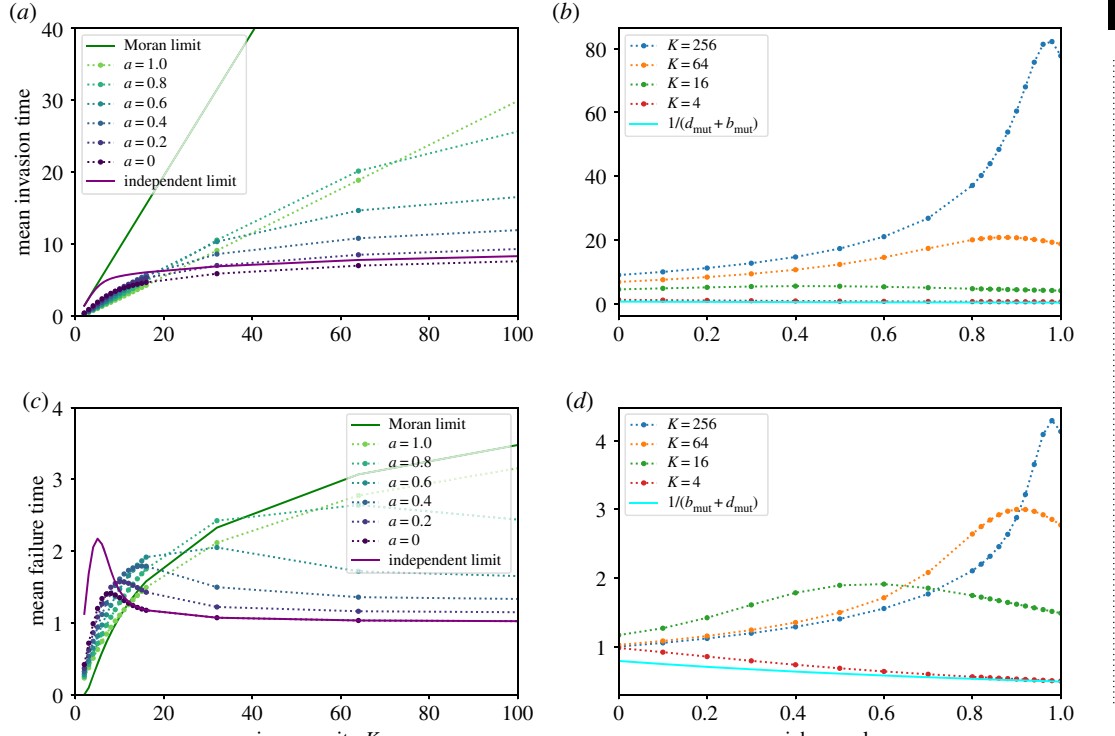

**Figure 5.** Mean time of a successful or failed invasion attempt. (a) Dotted lines show the numerical results of invasion times conditioned on success, from $a = 0$ at the bottom being mostly fastest to $a = 1$ being slowest. The solid green line shows for comparison the predictions of the Moran model in the complete niche overlap limit, $a = 1$; see text. The solid purple line corresponds to the solution of independent stochastic logistic species, $a = 0$, and overestimates the time at small $K$ but fares better as $K$ increases. (b) The red dotted line shows the successful invasion times for carrying capacity $K = 4$, and successive lines are at larger system sizes, up to $K = 256$. The cyan line is $1/(b_{mut} + d_{mut})$ and matches with numerical results at small $K$. (c,d) Same as upper panels, but for the mean time conditioned on a failed invasion attempt.

effect on the eventual outcome of the system, but a small change in the niche overlap between the invader and the established species leads to a large change in the probability of invasion success. For small $K$, either invasion or extinction typically occurs after only a small number of steps. The invasion probability in this limit is dominated by the probability that the lone mutant reproduces before it dies, namely $b_{mut}/b_{mut} + d_{mut} = K/(K(1 + a) + 1 − a)$, as shown in black solid line in figure 4b.

Figure 5a shows the dependence of the mean time to successful invasion, $\tau_s$, on $K$ and $a$. Increasing $K$ can have potentially contradictory effects on the invasion time, as it increases the number of births before a successful invasion on the one hand, while increasing the steepness of the potential landscape and therefore the bias towards invasion on the other. Nevertheless, the invasion time is a monotonically increasing function of $K$ for all values of $a$. In the complete niche overlap limit $a = 1$, the invasion time scales linearly with the carrying capacity $K$, as expected from the predictions of the Moran model, $\tau_s = \Delta t\ K^2(K − 1)\ln(K/(K − 1))$ with $\Delta t \simeq 1/K$, as explained above. The quantitative discrepancy arises from the breakdown of the $\Delta t \simeq 1/K$ approximation off the Moran line. In the opposite limit of non-interacting species, $a = 0$, the invading mutant follows the dynamics of a single logistic system with the carrying capacity $K$, resulting in the invasion time that grows approximately logarithmically with the system size, as shown in figure 5a,c as a solid purple line (see also electronic supplementary material). For all values $0 \leq a < 1$, the invasion time scales sublinearly with the carrying capacity, indicating that successful invasions occur relatively quickly, even when close to complete niche overlap, where the invading mutant strongly competes against the stable species.

By contrast, the failed invasion time, shown in figure 5c, is non-monotonic in $K$. The analytical approximations of the Moran model and of the two independent one-dimensional stochastic logistic systems recover the qualitative dependence of the failed invasion time on $K$ at high and low niche overlap, respectively. All failed invasion times are fast, with the greatest scaling being that of the Moran limit. For $a < 1$, these failed invasion attempts approach a constant timescale at large $K$.

The dependence of the time of an attempted invasion (both for successful and failed ones) on the niche overlap $a$ is different for small and large $K$, as shown in figure 5b,d. For small $K$, both $\tau_s$ and $\tau_f$ are monotonically decreasing functions of $a$, with the Moran limit having the shortest conditional times. In this regime, the extinction or fixation already occurs after just a few steps, and its timescale is determined by the slowest steps, namely the mutant birth and death. Thus $\tau \approx 1/(b_{\mathrm{mut}} + d_{\mathrm{mut}}) = K/(K + 1 + a(K - 1))$, as shown in the figures as the solid cyan line. By contrast, at large $K$, the invasion time is a non-monotonic function of the niche overlap, increasing at small $a$ and decreasing at large $a$. This behaviour stems from the conflicting effect of the increase in niche overlap: on the one hand, increasing $a$ brings the fixed point closer to the initial condition of one invader, suggesting a shorter timescale; on the other hand, it also makes the two species more similar, increasing the competition that hinders the invasion.

# 4. Discussion

Maintenance of species biodiversity in many biological communities is still incompletely understood. The classical idea of competitive exclusion postulates that ultimately only one species should exist in an ecological niche, excluding all others. Although the notion of an ecological niche has eluded precise definition, it is commonly related to the limiting factors that constrain or affect the population growth and death. In the simplest case, one factor corresponds to one niche, which supports one species, although a combination of factors may also define a niche, as discussed above. The competitive exclusion picture has encountered long-standing challenges as exemplified by the classical 'paradox of the plankton' [4,35] in which many species of plankton seem to cohabit the same niche; in many other ecosystems the biodiversity is also higher than appears to be possible from the apparent number of niches [31,35,39,114,115].

Competitive exclusion-like phenomena appear in a number of popular mathematical models, for instance in the competition regime of the deterministic LV model, whose extensive use as a toy model enables a mathematical definition of the niche overlap between competing populations [26–29]. Another classical paradigm of fixation within an ecological niche is the Moran model (and the closely related Fisher–Wright, Kimura and Hubbell models) that underlies a number of modern neutral theories of biodiversity [19,20,31,42–45]. Unlike the deterministic models, in the Moran model fixation does not rely on the deterministic exclusion of one species by the other, like in the strong competition and bistable regimes of the stability diagram figure 1 of the electronic supplementary material. Recently, the connection between deterministic models of the LV type and stochastic models of the Moran type has accrued renewed interest because of new focus on the stochastic dynamics of the microbiome, immune system, and cancer progression [13,25,43,49,56,57,73,89].

Much of the recent studies of these systems employed various approximations, such as the Fokker–Planck approximation [38,43,54,56,57], WKB approximation [55,74] or game-theoretic [49] approach. The results of these approximations typically differ from the exact solution of the master equation, used in this paper, especially for small population sizes [62,74,85,88,89]. In this paper, we have interrogated stochastic dynamics of a system of two competing species using a numerically arbitrarily accurate method based on the first passage formalism in the master equation description. The algorithmic complexity of this method scales algebraically with the population size rather than with the exponential scaling of the fixation time (as is the case with the Gillespie algorithm [93]). Our approach captures both the exponential terms and the algebraic prefactors in the fixation/extinction times for all population sizes and enables us to rigorously investigate the effect of niche overlap between the species on the transition between the known limits of the effectively stable and the stochastically unstable regimes of population diversity.

Stochastic fluctuations allow the system to escape from the deterministic coexistence fixed point towards fixation. If the escape time is exponential in the (typically large) system size, in practice it implies effective coexistence of the two species around their deterministic coexistence point. If the escape time is algebraic in $K$, as in the degenerate niche overlap case (closely related to the classical Moran model), the system may fixate on biological timescales [42,107]. Nevertheless, for biological systems with small characteristic population sizes, exponential scaling may not dominate the fixation time, and the algebraic prefactors dictate the behaviour of the system (see the electronic supplementary material for more details).

The transition from the exponential scaling of the effective coexistence time to the rapid stochastic fixation in the Moran limit is governed by the niche overlap parameter. We find that the fixation time

is exponential in the system size unless the two species occupy exactly the same niche. The numerical factor in the exponential is highly sensitive to the value of the niche overlap, and smoothly decays to zero in the complete niche overlap case. The implication is that two species will effectively coexist unless they have exactly the same niches.

Our fixation time results can be understood by noticing that the escape from a deterministically stable coexistence fixed point can be likened to Kramers' escape from a pseudo-potential well [54,62,92,116], where the mean transition time grows exponentially with well depth [62]. Approximating the steady-state probability with a Gaussian distribution shows that this well depth is proportional to $K(1-a)$ and disappears at $a = 1$. With complete niche overlap, the system develops a 'soft' marginally stable direction along the Moran line that enables algebraically fast escape towards fixation [54,57,90]. Similar to the exponential term, the exponent of the algebraic prefactor is also a function of the niche overlap and smoothly varies from $-1$ in the independent regime of non-overlapping niches to $+1$ in the Moran limit.

The fixation times of two coexisting species, discussed above, determine the timescales over which the stability of the mixed populations can be destroyed by stochastic fluctuations. Similarly, the times of successful and failed invasions into a stable population set the timescales of the expected transient coexistence in the case of an influx of invaders, arising from mutation, speciation or immigration. The probability of invasion strongly depends on the niche overlap, and for large $K$ decreases monotonically as $1-a$ with the increase in niche overlap. By contrast, the effect of the carrying capacity on the invasion probability is negligible in the large $K$ limit.

The mean times of both successful and failed invasions are relatively fast in all regimes, and scale linearly or sublinearly with the system size $K$. Even for high niche overlap, where the invasion probability is low due to the strong competition between the species, those invaders who do succeed invade in a reasonably fast time. In this high competition regime, the times of the failed invasions are particularly salient because they set the timescales for transient species diversity. If the influx of invaders is slower than the mean time of their failed invasion attempts, most of the time the system will contain only one settled species, with rare 'blips' corresponding to the appearance and quick extinction of the invader. On the other hand, if individual invaders arrive faster than the typical times of extinction of the previous invasion attempt, the new system will exhibit transient coexistence between the settled species and multiple invader strains, determined by the balance of the mean failure time and the rate of invasion [26,31,35,108].

The weaker dependence of the invasion times on the population size and the niche overlap, when compared with the escape times of a stably coexisting system to fixation, implies that the transient coexistence is expected to be much less sensitive to the niche overlap and the population size than the steady-state coexistence. Of note, both niche overlap and the population size can have contradictory effects on the invasion times (as discussed in §3) resulting in a non-monotonic dependence of the times of both successful and failed invasions on these parameters.

Our results indicate that the niche overlap between two species reflecting the similarity (or the divergence) in how they interact with their shared environment, is of critical importance in determining stability of mixed populations. This has important implications for understanding the long-term population diversity in many systems, such as human microbiota in health and disease [2,3,12], industrial microbiota used in fermented products [16], and evolutionary phylogeny inference algorithms [14,15]. Our results serve as a neutral model base for problems such as maintenance of drug resistance plasmids in bacteria [17] or strain survival in cancer progression [13]. The theoretical results can also be tested and extended based on experiments in more controlled environments, such as the gut microbiome of a *C. elegans* [73], or in microfluidic devices [117].

The results of the paper can also inform investigations of multi-species systems [24–26,30–32,35–38,69,90,118,119]. It is helpful to focus on one of the species within a large community of different species, which experiences a distribution of niche overlaps with other species in the system. Since extinction times away from complete niche overlap scale exponentially in the system size, the extinction of the given species is likely to be dominated by the shortest timescale, imposed by whichever other species has the greatest niche overlap with it, excluding it most strongly. Therefore, species extinctions are expected to occur more frequently in communities where the probability of high niche overlap between the species is high—for instance for distributions with high mean overlap, or high variance in the niche overlap distribution. These qualitative predictions compare favourably with the work of Capitán *et al.* [30], which shows that increasing the niche overlap in a multi-species system results on average in a smaller number of coexisting species, decreasing biodiversity. The probability of invasion into a multi-species system is more complicated than the coexistence

considerations, because, unlike the fixation times, invasion times do not scale exponentially with the system size, so no single pairwise interaction will dominate the timescale. These directions are beyond of the scope of the present work and will be studied elsewhere.

Data accessibility. The data can be generated using C++ code found at github.com/ma2canada/phd-code or doi.org/10.5061/dryad.t76hdr7xg [120].

Authors' contributions. M.B. and A.Z. designed the research, M.B. performed the research, M.B. and A.Z. wrote the paper. All authors gave final approval for publication.

Competing interests. The authors declare no competing interests.

Funding. The authors acknowledge partial support from the Natural Sciences and Engineering Research Council of Canada (NSERC); discovery grant no. RGPIN-2016-06591.

Acknowledgements. The authors are thankful to Sid Goyal, Jeremy Rothschild and Matt Smart for their helpful discussions.

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
