## [Reviewer comments · Royal Society Open Science]

Review History

Decision letter (RSOS-192181.R0)

08-Jan-2020

Dear Mr Badali

On behalf of the Editors, I am pleased to inform you that your Manuscript RSOS-192181 entitled "Effects of niche overlap on co-existence, fixation, and invasion in a population of two interacting species" has been accepted for publication in Royal Society Open Science subject to minor revision in accordance with the referee suggestions. Please find the referees' comments at the end of this email.

The reviewers and handling editors have recommended publication, but also suggest some minor revisions to your manuscript. Therefore, I invite you to respond to the comments and revise your manuscript.

- Ethics statement

- Data accessibility

If you wish to submit your supporting data or code to Dryad (<http://datadryad.org/>), or modify your current submission to dryad, please use the following link:
<http://datadryad.org/submit?journalID=RSOS&manu=RSOS-192181>

- Competing interests

- Authors' contributions

- Acknowledgements

- Funding statement

Because the schedule for publication is very tight, it is a condition of publication that you submit the revised version of your manuscript before 17-Jan-2020. Please note that the revision deadline will expire at 00.00am on this date. If you do not think you will be able to meet this date please let me know immediately.

To revise your manuscript, log into <https://mc.manuscriptcentral.com/rsos> and enter your

Author Centre, where you will find your manuscript title listed under "Manuscripts with Decisions". Under "Actions," click on "Create a Revision." You will be unable to make your revisions on the originally submitted version of the manuscript. Instead, revise your manuscript and upload a new version through your Author Centre.

If your manuscript is newly submitted and subsequently accepted for publication, you will be asked to pay the article processing charge, unless you request a waiver and this is approved by Royal Society Publishing. You can find out more about the charges at <https://royalsocietypublishing.org/rsos/charges>. Should you have any queries, please contact opscience@royalsociety.org.

on behalf of Dr Pietro Cicuta (Associate Editor) and Pietro Cicuta (Subject Editor)
openscience@royalsociety.org

Associate Editor Comments to Author (Dr Pietro Cicuta):

Associate Editor

Comments to the Author:

Reviews from Interface are detailed and positive for RSOS. Reviewer 3 has a series of outstanding "other comments" that should be addressed without further peer review.

Author's Response to Decision Letter for (RSOS-192181.R0)

See Appendix A.

Decision letter (RSOS-192181.R1)

21-Jan-2020

Dear Mr Badali,

It is a pleasure to accept your manuscript entitled "Effects of niche overlap on co-existence, fixation, and invasion in a population of two interacting species" in its current form for publication in Royal Society Open Science. The comments of the reviewer(s) who reviewed your manuscript are included at the foot of this letter.

Due to rapid publication and an extremely tight schedule, if comments are not received, your

paper may experience a delay in publication. Royal Society Open Science operates under a continuous publication model. Your article will be published straight into the next open issue and this will be the final version of the paper. As such, it can be cited immediately by other researchers. As the issue version of your paper will be the only version to be published I would advise you to check your proofs thoroughly as changes cannot be made once the paper is published.

on behalf of Dr Pietro Cicuta (Associate Editor) and Pietro Cicuta (Subject Editor)
openscience@royalsociety.org

Associate Editor Comments to Author (Dr Pietro Cicuta):

Associate Editor: 1

Comments to the Author:

(There are no comments.)

Reviewer comments to Author:

Appendix A

Response to the technical comments by Referee 2

In this revision, the authors have addressed my technical concerns in the previous version (I have a few comments below) [...].

We thank the referee for the extremely careful reading and overall positive opinion. The responses to the Referee's queries (in blue) are below (in black).

Other comments

p. 2, line 31 I think "independent" is a more appropriate term than "weak competition", as the latter implies that there is some competition. It seems to me that the term weak competition would be better reserved for the regime considered in [4, 5], in which intrinsic birth and death rates differ by order $O(1/K)$, whereas competition-induced mortality is of order $O(1/K^2)$; in this case, there is another diffusive scaling, not limited to a submanifold, the logistic Feller Diffusion.

We have changed the wording accordingly.

p. 6, line 21 Some more care is required in the discussion of the extinction time distribution here; in general, it will depend on the initial state of the process and is only asymptotically exponential (see e.g., [1])

We have changed the wording accordingly.

p. 6, lines 31-32 Please define ΔK before using it.

We assume that the referee meant Δt . We have rephrased the sentence accordingly.

p. 6, lines 41-42. The WKB ansatz was applied to analyze extinction probabilities, quasi-stationary distributions and the quasi-potential in a model of competing species (with xed points) several decades earlier in [7, 8].

We assume the referee meant references [9,10], which have now been added.

p. 7, line 33. Please be more precise about $a \leq 1$. One would expect diffusive behaviour on a sub-manifold similar to the Moran limit when $1 - a = O(1/K)$ ([2, 6])

We have clarified the meaning of the sentence.

p. 37, lines 37-38 How small is K required to see dominance of the algebraic terms? Is this a biologically realistic carrying capacity?

We assume the referee refers to p.7, lines 37-38. We have added a clarification in the Supplementary Information referring to this point.

p. 9, lines 42-44. That the invasion probability is non-monotonic in K admits a relatively simple intuition: increasing K simultaneously makes invasion more difficult (it becomes harder to reach $K/2$ as K increases, but this effect becomes unimportant with increasing K , see below) and less difficult (competition, and thus individual mortality decreases with K). Indeed, given the current definition of invasion as reaching $K/2$, one has the trivial result that invasion occurs with probability 1 at $K = 2$.

The referee is correct. We have added an explanation of the non-monotonicity in the text.

p. 9, lines 52-53 The relative independence of the invasion probability on K for large K shouldn't be unexpected: as K becomes large, as the authors observe, the invading population is well approximated by a birth and death process with rates depending only on the density of the resident population (and thus independent of K); this approximation fails only when the invading population consists of $O(1/K)$ individuals, in which case it is well approximated by the deterministic (2.1) (see [?] and e.g., [?] for a discussion of how the two approximations overlap).

The referee is correct, and this point is indeed explained in the text. We have removed the word "unexpectedly" from the sentence. Unfortunately, we are not able to cite suggested references as their numbers do not appear in the referee report.

p. 12, lines 11-12 What do the authors mean when they say "in the Moran model fixation does not rely on deterministic competition for space [...] but arises from stochastic demographic noise"? The Moran model, and the Wright-Fisher model that preceded it explicitly included a fixed population size as a means of modelling competition for space (as opposed to the branching process models of Haldane, which modelled demographic stochasticity without competition for resources).

We apologize for the confusion. We mean that in the Moran model, the fixation does not occur due to the deterministic exclusion of one species by the other, like in the strong competition and bistable regimes of the stability diagram of the Figure 1 of the Supplementary Information. We have clarified that sentence accordingly.

References

- [1] H. Andersson and B. Djehiche. A threshold limit theorem for the stochastic logistic epidemic. *J. Appl. Prob.*, 35:662{670, 1998.
- [2] N. Champagnat. A microscopic interpretation for adaptive dynamics trait substitution sequence models. *Stochastic Processes Appl.*, 116(8):1127{1160, 2006.
- [3] George W. A. Constable and Alan J. McKane. Models of genetic drift as limiting forms of the lotka-volterra competition model. *Phys. Rev. Lett.*, 114:038101, Jan 2015.

- [4] M. I. Fredlin, , and A. D. Wentzell. Random perturbations of dynamical systems. Springer-Verlag, New York, 1984.
- [5] T. G. Kurtz. Solutions of ordinary differential equations as limits of pure jump Markov processes. *J. Appl. Prob.*, 7(1):49{58, 1970.
- [6] A. Lambert. The branching process with logistic growth. *Ann. Appl. Probab.*, 15(2):1506-1535,2005.
- [7] A. Lambert. Probability of Fixation under weak selection: A branching process unifying approach. *Theor. Popul. Biol.*, 69(4):419 { 441, 2006.
- [8] T. L. Parsons and T. Rogers. Dimension reduction for stochastic dynamical systems forced onto a manifold by large drift: a constructive approach with examples from theoretical biology. *J. Phys. A: Math. Theor.*, 50(41):415601, 2017.
- [9] H. Roozen. Equilibrium and extinction in stochastic population dynamics. *Bull. Math. Biol.*, 49(6):671{696, Nov 1987.
- [10] H. Roozen. An asymptotic solution to a two-dimensional exit problem arising in population dynamics. *SIAM J. Appl. Math.*, 49(6):1973{1810, 1989.